# Artificial Intelligence & Tissue Biomarkers: Advantages, Risks and Perspectives for Pathology

**DOI:** 10.3390/cells10040787

**Published:** 2021-04-02

**Authors:** Cesare Lancellotti, Pierandrea Cancian, Victor Savevski, Soumya Rupa Reddy Kotha, Filippo Fraggetta, Paolo Graziano, Luca Di Tommaso

**Affiliations:** 1Department of Biomedical Sciences, Humanitas University, Via Rita Levi Montalcini 4, 20090 Pieve Emanuele, Milan, Italy; cesare.lancellotti@humanitas.it (C.L.); soumya.kotha@st.hunimed.eu (S.R.R.K.); 2Pathology Unit, Humanitas Research Hospital, Via Manzoni 56, 20089 Rozzano, Milan, Italy; 3Artificial Intelligence Center IRCCS Humanitas Research Hospital, Via Manzoni 56, 20089 Rozzano, Milan, Italy; pierandrea.cancian@humanitas.it (P.C.); victor.savevski@humanitas.it (V.S.); 4Department of Pathology, Cannizzaro Hospital, 95021 Catania, Italy; filippofra@hotmail.com; 5Pathology Unit, Fondazione IRCCS Casa Sollievo della Sofferenza, 71013 San Giovanni Rotondo, FG, Italy; p.graziano@operapadrepio.it

**Keywords:** biomarker, artificial intelligence, pathology, personalized medicine

## Abstract

Tissue Biomarkers are information written in the tissue and used in Pathology to recognize specific subsets of patients with diagnostic, prognostic or predictive purposes, thus representing the key elements of Personalized Medicine. The advent of Artificial Intelligence (AI) promises to further reinforce the role of Pathology in the scenario of Personalized Medicine: AI-based devices are expected to standardize the evaluation of tissue biomarkers and also to discover novel information, which would otherwise be ignored by human review, and use them to make specific predictions. In this review we will present how AI has been used to support Tissue Biomarkers evaluation in the specific field of Pathology, give an insight to the intriguing field of AI-based biomarkers and discuss possible advantages, risk and perspectives for Pathology.

## 1. Introduction

Pathologists base their routine clinical practice on the recognition, semi-quantification and integration of morphological patterns according to predefined criteria dictated by the clinical context. These data are then classified and summarized in the histopathological report. The natural differences in visual perception, data integration and judgement between each pathologist makes Pathology a subjective discipline. The first efforts to extract objective measures from microscopic images were produced in cytology. The introduction of scanners in the 1990s allowed the creation of digitized images of tissue slides (Whole Slide Images, WSI) and led to a growing interest in the application of Artificial Intelligence (AI) in Pathology, which previously happened in the field of Radiology [1,2,3]. Machine Learning (ML) was the first AI technique applied in the field of Pathology. ML-algorithms were based on engineering specific morphological features used by pathologists for a specific task, for example cell size, nuclear shape and cytoplasm texture to discriminate between benign and malignant tumors. The main limitation of ML was that annotation of specific features was timeconsuming and sometimes anchored to the original problem domain (not scalable to other problems). The introduction of Deep Learning (DL) boosted the explosion of AI applied to histopathology. Indeed DL approaches can learn directly from raw data (WSI) and do not rely on the effort to engineer a feature anchored to the problem as in ML. Nonetheless, even if DL does not need pre-existing assumptions, raw data might need a certain degree of control. DL approaches can be further subdivided into strongly supervised, weakly supervised or unsupervised. In strongly supervised DL, several patches extracted from WSI should be labelled by pathologist with the class that the model is intended to predict. In weakly supervised approaches annotations are made at image level (each WSI is labelled with a specific class). In these models, a single label of interest might be used to develop an algorithm: the presence or absence of tumor (the most common); the presence of a somatic mutation; the clinical outcome. While easier to be labelled, series evaluated by weakly supervised DL models usually need to be larger than those used for strictly supervised approaches. As regard to the dimension of the dataset it is interesting to observe that convolutional neural network (CNN) can be trained on a source task and then be reused on a different target task. This technique, known as transfer learning, can be extremely useful when the data for the target task is scarce but a larger dataset is available to train the source task [4,5]. Finally, in unsupervised DL, the learning examples are provided with no associated labels. This approach, which to date has been applied in a very limited number of examples [6] is an active area of machine learning research.

The term Digital Pathology (DP) encompasses all the digital technologies related to the introduction of WSI that allow improvements and innovations in the workflow of a Pathology Department [7]. AI-based tools are part of these technologies: starting from WSI they promise to improve pathologists’ activity to recognize, quantify and integrate information written in the tissue (tissue biomarker) for diagnostic, prognostic and predictive purposes. In order to develop such AI-based tools, researchers need specific software to work on WSI. These include color normalization, focus quality assessment, standard tiles extraction, object classification, region segmentation and counting. Table 1 lists some among the most popular software used to these aims. A list far to be complete taking into consideration that the open source Github (https://github.com/, accessed on 26 February 2021) returns more than 100 software solutions tagged by “digital pathology” while a recent forecast for DP market (https://www.marketsandmarkets.com/Market-Reports/digital-pathology-market-844.html#:~:text=%5B210%20Pages%20Report%5D%20The%20global,13.8%25%20during%20the%20forecast%20period.&text=However%2C%20a%20lack%20of%20trained,growth%20in%20the%20coming%20years, accessed on 26 February 2021) catalogue more than 20 companies working on this topic. The platform BIII (https://biii.eu/, accessed on 26 February 2021) developed by the Network of European Bio-image Analyst NEUBIAS proposes to help the researcher to compare this plethora of solutions focusing on the problems the tools can solve rather than comparing their technical aspects. Finally, it should be remembered that AI-tools which are to be implemented in clinical practice are subject to strict regulation. The new Conformite Europeénne—in vitro diagnostic device regulation (CE-IVDR) will significantly impact this process in EU from 2022. Table 2 lists CE approved AI-based tools and their application field.

## 2. Tissue Biomarkers and Artificial Intelligence

In the setting of Personalized Medicine (PM) a biomarker is every piece of information used to recognize a specific subset of a larger population with diagnostic, prognostic and/or predictive purposes. Tissue biomarkers are information written in the tissue, which can be interpreted by properly only by Pathology. Histotype, Grade and Stage of malignant tumors are “classic” tissue biomarkers: recognizing patients affected by early stage disease and with specific favorable histotype, represent the origin of PM. Estrogen and progesterone receptors, Ki67 and HER2/neu, aka the biological profile of breast cancer, were the first examples, in the 2000s, of “new” tissue biomarkers: the expression of these phenotypical indicators allowed to select properly medical treatment and predict outcome of patients affected by breast cancer. In the last two decades, several other biomarkers have been discovered at different deepness (cellular, subcellular, molecular) in distinctive cancer population (neoplastic cells, cancer-associated immune cells, etc.). The increasing number of tissue biomarkers and the complexity of their evaluations, strongly encourages the use of AI based tools in the process of evaluation, to reinforce the role of Pathology in PM. [8]. Moreover, AI algorithms have been used to discover information which are ignored by human review of an H/E image and use them to make specific prediction as shown in Figure 1. These AI-based biomarkers include the prediction of treatment response [9,10], somatic mutations [11], or patient survival [12,13]. In the following sections, we will present how AI has been used to support tissue biomarkers evaluation in specific field of Pathology; also, we will give an insight to the intriguing field of AI-based biomarkers.

### 2.1. AI in Breast Pathology

Several AI-based algorithms have been proposed in the field of breast digital Pathology. Osareh et al. introduced a model based on 10 cellular features to accurately distinguish between benign and malignant lesions [14]. Han et al. [15] trained a classifier which distinguished between benign and malignant breast tumors with 93.2% accuracy. Cruz-Roa et al. [16] used manually annotated regions obtained from 400 slides as training sets and 200 slides with similar annotations from TCGA to develop a DL-algorithm able to recognize invasive ductal carcinoma with 75.8% accuracy. AI models have also been applied to distinguish in situ from invasive breast cancer [17]. AI solutions designed to detect nodal metastasis of breast cancer outperformed a panel of pathologists in a diagnostic simulation: the best performance achieve by a pathologist showed an AUC of 0.88 as compared to the 0.99 AUC of the best algorithm [18]. Other studies showed that the average review time was significantly shorter with AI-assistance in WSI of lymph node without metastases (1.2 times faster) and with micrometastases (1.9 times faster) [19,20]. In addition to tumor identification, several AI algorithms have been proposed for breast cancer grading: most focused on mitosis detection while those for the evaluation of tubular formation and nuclear grade are still to be developed. Interestingly, in mitosis detection tasks AI got close but did not reach pathologists’ analysis [21]. AI methods, in particular ML-based, have been proposed to classify histologic subtypes of breast cancer [22]. The quantification of the biological profile of breast cancer, namely the evaluation of ER, PR and HER2, was an early application of AI in digital breast cancer Pathology [19,20,21,22,23,24]. Rexhepaj et al. [23] used a detection algorithm to quantify ER and PR expression and found a correlation >90% between manual and algorithmic quantification. Skaland et al. [24] found 100% concordance between the algorithm’s prediction and HER2 status assessment. Interestingly, the transfer learning approach has been tested in the field of breast cancer and proved remarkable performance as compared to CNN with full training [25,26].

Several studies also explored the prediction of clinical features directly from H&E slides bypassing immunohistochemical staining and molecular characterization, thus producing AI-based biomarkers. Couture et al. [21] tested both ML- and DL- models to predict the molecular features from H/E and reported a final accuracy of 84% for the prediction of ER status. Shamai et al. [27] proposed a system to predict the statuses of 19 biomarkers, including ER and PR and reported a 92% accuracy for ER status within the subgroup of high-confidence cases. Rishi et al. [28] trained the algorithm to learn the morphological differences among different tumors assuming that this will implicitly teach it about the biologic differences between them. The features the network learned, called “fingerprints,” enabled the determination of ER, PR and Her2 status from whole slide H&E images with 0.89 AUC (ER), 0.81 AUC (PR) and 0.79 AUC (Her2) on a large, independent test set (*n* = 2531). Lu et al. [29] proposed an algorithm able to separate, on the bases of nuclear shape and orientation of neoplastic cells, ER-positive breast cancer patients into short-term (<10 years) and long-term (>10 years) survival. Romo-Bucheli et al. [30] developed an algorithm to compute nuclear features to predict Oncotype DX risk categories. The model developed by Whitney et al. [31], using nuclear shape, texture and architecture, was able to predict the risk of recurrence in ER-positive breast tumors as compared to Oncotype DX.

### 2.2. AI in Prostate Pathology

Contrary to breast Pathology, where several tissue biomarkers have been and are currently being investigated by ML and DL tools, studies on AI application to the field of prostate Pathology are limited to the detection and the grading of prostatic adenocarcinoma. To better understand this aspect is important to remember that in the USA about one in nine men will develop prostatic cancer and that most of them will experience a 12/18-core-biopsy of the prostate at least once. The huge number of slides generated with this procedure, the dramatic shortage of pathologists [32] and a risk of false negative up to 8% [33], likely explain the great interest in detection and grading of prostatic adenocarcinoma using AI. This is also confirmed by the increasing number of papers on the topic. Nagpal et al. [34] developed a DL algorithm using as reference a group of expert uropathologists, and reported that their model performed significantly better than general pathologists on tumor grading in prostatic biopsy (71.7% versus 58.0%; *p* < 0.001). Interestingly enough, the AI algorithm achieved the same results of general pathologists in the recognizing the presence of tumor (94.3% for the AI and 94.7% for general pathologists). The same group developed a model which assigned Gleason scores on radical prostatectomy specimens with an accuracy of 0.70, as opposed to a mean accuracy of 0.61 of general pathologists [35]. Ström P et al. [36] trained a DL method on WSIs obtained from prostatic biopsy of the Swedish population-based study STHLM3. The algorithm achieved an AUC of 0.986 for distinguishing between benign and malignant biopsy; of 0.87 for cancer length prediction; and 0.62 for assigning Gleason grades, within the range of the corresponding values for the expert pathologists (0.60–0.73). Bulten W et al. [37] reported about an AI-solution reaching an AUC of 0.99 in the distinction between benign versus malignant foci; of 0.98 for grade group of 2 or more; and of 0.974 for grade group of 3 or more. Moreover, the authors also observed that the AI system scored higher than a panel of pathologists, outperforming 10 out of 15 of them. AI systems could potentially assist pathologists not only in screening biopsies, cancer grading and measurements of tumor volume/percentage but also in providing a second-read opportunity. Pantanowitz et al. [38], in a prospective series of 941 cases, reported at least one case of cancer detected by AI which was missed by the pathologist on initial review.

Despite being focused on only a few aspects, it is likely that AI solutions in prostatic cancer will be the first to be used in the routine digital pathology practice. Indeed, during the last few months two softwares, Paige Prostate by PaigeAI and Galen™ Prostate by Ibex Medical Analytics, received the CE Mark for supporting pathologists in the identification of prostate cancer on core needle biopsies.

### 2.3. AI in Lung Pathology

Attempts have been made to use AI solutions in lung cancer pathology and most of these focused on the possibility to predict from H/E clinical information missed by the human eye, namely AI-based biomarkers. Yu et al. [39] trained a ML solution on a 9879 image features obtained from 2186 H/E WSI of lung adenocarcinoma (ADK) and squamous cell carcinoma (SqCC) patients from TCGA and reported that the top features selected (mostly regarding nuclear features of neoplastic cells) distinguished, in validation cohort, shorter- from longer- term survivors in stage I ADK (P 0.003) and SqCC (P 0.023). Coudray et al. [11] trained a DL model to predict the most frequent gene mutations of lung ADK from H/E images and reported for 6 genes (including EGFR, KRAS and TP53) an AUC ranging from 0.733 to 0.856. Sha et al. [40] developed a DL model to predict PD-L1 status in non-small cell lung cancer (NSCLC) from H/E slides and reported it was significantly predictive in ADK (AUC 0.85) but not in SqCC (AUC 0.64). They also observed that model remained effective with different PD-L1 cutoff and when simulating pathologist disagreement. AI solutions have been proposed to predict responsiveness to nivolumab in advanced stage NSCLC patients according to nuclear features [41] or the spatial arrangement of tumor infiltrating lymphocytes (TIL) [42].

Some studies also explored how to use AI solution to improve the evaluation of tissue biomarkers. Coundray et al. [11] trained their AI platform to classify lung cancer into ADK and SqCC and reported a good performance (AUC: 0.86–0.97) when it was tested on in three independent cohorts. Algorithms for recognizing lung adenocarcinoma subtype have been developed by some groups [42,43]. The main challenge in this area is the reliability of the annotations used since they can vary among pathologists or among institutions. Wei et al. [44] reported that the agreement between the trained AI and a pathologist had a similar low value to that observed between pathologists who annotated original data. This is understandable as AI internalizes any bias can be influenced by discordant annotation labels during training. Althammer S et al. [45] using customized algorithm reported that PD-L1+ and CD8+ cell densities were related to response to anti PD-L1 therapy. A DL-based approach [46] has also been used to score PD-L1 expression in images of NSCLC biopsy samples. This approach helped to minimize the number of pathologist annotations necessary and thus compensates for the lack of tissue available in a biopsy specimen.

### 2.4. AI in Colon Pathology

There are few studies exploring AI solutions in the field of colorectal cancer pathology. Most of these focused on gland segmentation, i.e., to recognize the structure, shape and size of glands, a crucial task for grading purpose. A specific challenge on this topic, GLAs 2015, rewarded the CUMedVision model [47]. Other studies investigated the possible application of AI solution to classification task, such as the distinction between colorectal polyps and histologic subtypes of adenocarcinoma achieving interesting results [48,49,50,51].

More recently, two studies proposed AI-based biomarkers as prognostic and predictive of colorectal cancer (CCR). Bychkov et al. [52] developed a DL model to predict the risk 5-years CCR recurrence (low versus high, based on retrospective data) using images of H&E-stained Tissue Micro Array (TMA) specimens. The digital risk score generated by the model outperformed (AUC 0.69) the prediction based on grading and/or staging established by expert pathologist on both TMA spot (AUC 0.58) and whole-slide level (AUC 0.57). A DL approach has also been suggested to predict MicroSatellite Instability (MSI), an actionable molecular phenotype which is regularly tested for in the clinical laboratory, from H&E images [53]. The authors first developed a tumor detector with an AUC > 0.99; then trained another AI model to classify MSI versus microsatellite stability (MSS) in large patient cohorts from TCGA both from formalin-fixed and fresh tissue. Then combining the two models the authors proved robust performance across a range of human tumors (AUCs for MSI detection were 0.81for CCR; 0.75 for endometrial cancer; and 0.69 for gastric cancer) and exceeded the previously reported performance of predicting molecular features from histology.

### 2.5. AI Solutions Independent of Cancer Cells

Geessink et al. [54] generated a DL model able to quantify the stromal component within the tumor and demonstrated that the tumor/stroma ratio was independently prognostic for DFS in colorectal cancer in a multivariable analysis incorporating clinicopathological factors. Kather et al. [9] used a DL model to generate a ‘deep stroma score’ and found it to be independently prognostic of RFS and OS in colorectal cancer. Beck et al. [55], extracted from WSIs of breast cancer 6642 features related to morphology as well as to spatial relationships and global image features of epithelial and stromal regions. They then used these features to train a prognostic model and found features extracted from the stromal compartment had a stronger prognostic value (*p* = 0.004) compared to features extracted from the epithelial compartment (*p* = 0.02).

The presence and organization of tumor infiltrating lymphocytes (TIL) impact on clinical outcome of several tumors. Accordingly, several AI solutions have been developed to explore TIL features. Saltz et al. [10], using a DL solution to detect and quantify the structure of TIL, found that this feature was prognostic of outcome for 13 different cancer subtypes in images from TCGA. Moreover, the integrated analysis of TIL maps and molecular data demonstrated that the local patterns and overall structural patterns of TILs are differentially represented amongst tumor types, immune subtypes and tumor molecular subtypes.

Yuan et al. [56] proposed a method to analyze the spatial distribution of lymphocytes among tumor cells in triple-negative breast cancer and found that the ratio of intratumoral lymphocytes to cancer cells was independent predictor of survival and correlated with the levels of cytotoxic T lymphocyte protein 4 (CTLA-4) expression. Heindl et al. [57] found that the spatial distribution of immune cells was also associated with late recurrence in ER-positive breast cancer. Corredor et al. [58] using an AI model which calculated the relationships between TILs proximally located to each other and between TILs and cancer cell, found that the spatial arrangement of clusters of TILs rather than TIL density alone, was strongly prognostic of recurrence risk in early stage NSCLC.

Finally, in a recent paper, H/E slides and RNA-Seq data of 28 different cancer types collected from TCGA were used to train a neural network to detect and predict which gene was the most likely to be involve in the specific type of cancer [59].

## 3. Artificial Intelligence in Pathology: Future Perspetives

The application of AI in the real clinical setting is still limited by several issues.

The low level of digitization likely represent the first critical issue. A recent survey in England revealed that an access to a complete DP workstation was available in less than 30% of Institutions; the most common applications were teaching, research and quality assurance while direct clinical use was less widespread with consultations outdating primary diagnosis [60]. Lack of robustness and general applicability is another restriction to the application of AI in the daily practice. Most of the available AI models have been trained on small data sets and can present a 20% drop of performance when applied in a setting different from where they had been originated. Dataset can be enlarged using specific technical solution such as the transfer learning approach [4,5]. Another possibility will be the development of open sources datasets such as those already hosted by the Cancer Genome Atlas, the Cancer Imaging Archives and Grand Challenges. Recently a call for a “*central repository to support the development of artificial intelligence tools*”, was proposed by the H2020 program IMI2-2019-18. This dataset aims to endorse WSI, molecular and clinical data and to serve as raw data for the scientific community. In addition to large series, another prerequisite to AI elaboration is well-annotated ground truth generated by expert pathologists with specific, time-consuming sessions. This is in striking contrast to one of the popular motivations for the introduction of AI in pathology, namely the shortage of pathologists. Technical solutions, such as the data augmentation [61], image synthesis [62] and the adoption of weakly supervised or unsupervised DL model, have been suggested or are actively explored to fix this paradox. Finally variations in staining procedures, tissue types and scanners might be relevant to obtain higher performing AI systems [63]. Low adherence amongst the use of the AI models by pathologists can be another source of limitation. In their clinical activity, skilled pathologists examine the slide in two steps: first a scanning magnification to understand the general context of the disease; then a more careful evaluation with progressively higher magnifications to prove their general impression. Most AI models are developed using smaller tiles, rather than entire WSI, as input data, missing the efficacy warranted by the dual approach of the pathologist. To avoid this drawback recent studies have suggested the introduction of networks trained with images obtained at different magnification [64,65]. A different technical solution was proposed by Lin et al. [66] who introduced in the neural network a further layer aimed to reconstruct the loss occurred in max pooling layers. Also the interpretation of AI decisions, sometimes referred to as the ‘black box’ problem [67], is a relevant concern to complete adoption of AI. To overcome this, it has been suggested to link the solution proposed by AI models to different type of visual maps describing the abundance and morphology of features (necrosis, pleomorphism, etc.) known by pathologists [68,69,70].

On the other hand, a full integration of AI in Pathology is likely to represent a milestone of digital health. The introduction of AI as a device assisting pathological diagnosis is expected to reduce the workload of pathologists; to help standardize the otherwise subjective diagnosis that can lead to suboptimal treatment of patients; to help discovery new perspectives in human biology, and progress on personalized diagnostics and patient care [71]. In the current state AI platforms are developed with different functions, requiring users to launch different software for each purpose or to repeatedly download and upload images. The development of a simplified user interface, either on the WSI viewer or on the LIS (laboratory integration system), is a key factor to the successful implementation of AI at clinical level. Moreover, the availability of platforms integrating different software solutions with multiple clinical data to suggest prognosis and/or the choice of therapy will be another substantial benefit of digitization and provide an additional sanity check on AI generated predictions. Another relevant aspect of progressive digitalization and introduction of AI in Pathology will be a more integrated approach with radiomic. The latter rests on the hypothesis that data derived from digital radiological images have a correlation with the underlying biological processes and that this correlation can be caught by AI better than the visual interpretation. Pioneering studies recently explored the associations between radiomic and its counterpart on digital Pathology images (i.e., pathomic) in lung and breast cancer and revealed promising preliminary results [72,73].

When such advanced (“next generation”) pathological diagnosis enter medical practice, it is likely that the demands of clinicians would not be satisfied with the level of current pathological diagnosis offered by pathologists using solely a microscope. Pathologists who reject digital Pathology and AI may face a diminished role in the future of Pathology practice.

## Figures and Tables

**Figure 1 cells-10-00787-f001:**
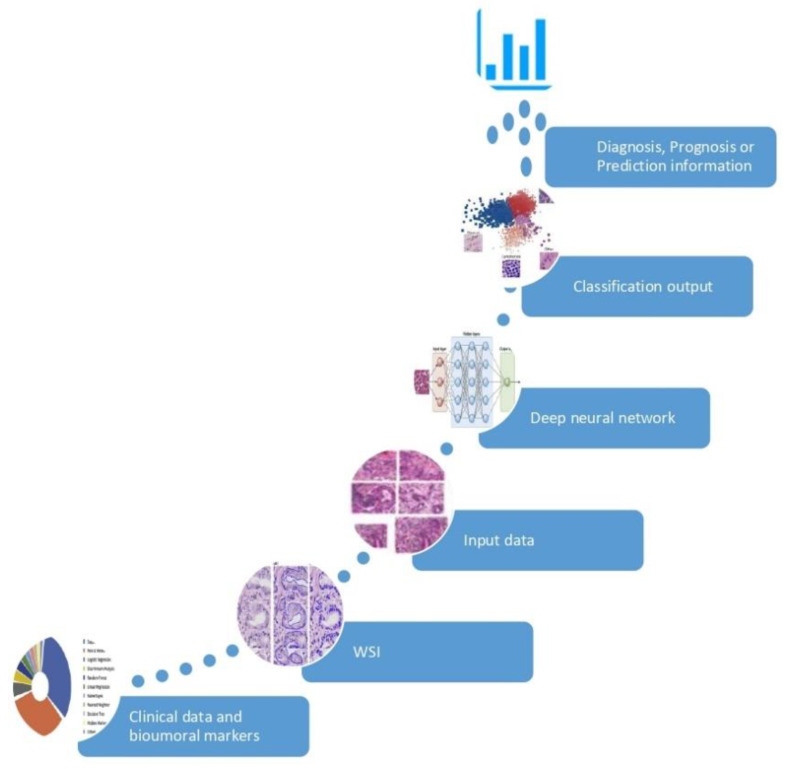
Development of an AI-based biomarker. The AI model is fed by input data (huge collection of clinical information and digital images) and learns the optimal feature to best separate the categories of interest, without pre-existing assumptions. The classification outcome returns an information of significant clinical impact in the diagnosis, prognosis or prediction.

**Table 1 cells-10-00787-t001:** List of most common open and commercially available AI software.

QuPath	https://qupath.github.io, accessed on 26 February 2021
HistoQC	https://github.com/choosehappy/HistoQC, accessed on 26 February 2021
ASAP	https://computationalpathologygroup.github.io/ASAP, accessed on 26 February 2021
PyHIST	https://github.com/manuel-munoz-aguirre/PyHIST, accessed on 26 February 2021
HistomicsTK	https://digitalslidearchive.github.io/HistomicsTK/, accessed on 26 February 2021
Histolab	https://histolab.readthedocs.io/en/latest/, accessed on 26 February 2021
PytorchDigitalPathology	https://github.com/CielAl/PytorchUnet, accessed on 26 February 2021
Visiopharm	https://visiopharm.com/, accessed on 26 February 2021
Paige	https://paige.ai/, accessed on 26 February 2021
Ibex	https://ibex-ai.com/, accessed on 26 February 2021
Aiforia	https://www.aiforia.com/, accessed on 26 February 2021
Proscia	https://proscia.com/, accessed on 26 February 2021

**Table 2 cells-10-00787-t002:** List of certified Artificial Intelligence (AI) solutions and clinical area of interest.

Name	Certification	Clinical Area
INFINY^®^	CE	Prostate cancer screening
GALEN™ PROSTATE	CE	Prostate cancer screening (1st READ) Prostate cancer quality control (2nd READ)
PAIGE PROSTATE CLINICAL	CE	Prostate cancer screening
PAIGE BREAST CLINICAL	CE	Breast cancer screening
Deep DX—PROSTATE pro	CE	Prostate cancer screening

Conformite Europeénne (CE) Marking is required for all in vitro diagnostic (IVD) devices sold in Europe. CE Marking indicates that an IVD device complies with the European In-Vitro Diagnostic Devices Directive (98/79/EC) and that the device may be legally commercialized in the European Union (EU).

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
