# Peer review of "Artificial Intelligence & Tissue Biomarkers: Advantages, Risks and Perspectives for Pathology"

_cells, 2021, doi:10.3390/cells10040787_

Round 1
Reviewer 1 Report
The paper by Cesare et al is well written and I only have a few suggestions:
The first part would benefit from an image showing the work flow of AI in pathology.
Any example of tissue images analyzed with AI would also benefit the paper.
The subtitle TISSUE-INDIPENDENT AI SOLUTIONS is misleading. The authors could maybe use AI SOLUTIONS INDEPENDENT OF CANCER CELLS
Reviewer 2 Report
The review is interesting and focussed on a quite hot topic; furthermore, it is sufficiently readable also by a non-specialistic audience and it is reasonably broad to cover several aspects of the investigated subject.
However, there are a number of issues that the authors need to address, and I briefly list them hereafter. Some important topics are only mentioned or even overlooked, while I reckon they deserve a deeper discussion, namely:
- Reproducibility
- Integration with radiomics approaches
- The transfer learning approach
- The use of smaller tiles as input data and the linked ML problem of information/data leakage
- A more organised discussion concerning the available software: see for instance PyHist, HistomicsTK, HistoQC, Histolab, deep-histopath, compay-syntax, py-wsi
- Reference list is rich, but some important papers have not been considered; hereafter I list some of them, but more should be added:.
- Shallu, R.M.: Breast cancer histology images classification: training from scratch or transfer learning? ICT Exp. 4(4), 247–254 (2018)
- Pan, X., et al.: Multi-task deep learning for fine-grained classification/grading in breast cancer histopathological images. In: Lu, H. (ed.) ISAIR 2018. SCI, vol. 810, pp. 85–95. Springer, Cham (2020).
- Deniz, E., Sengur, A., Kadiro glu, Z., Guo, Y., Bajaj, V., Budak, U ̈.: Transfer learning based histopathologic image classification for breast cancer detection. Health Inf. Sci. Syst. 6(1), 1–7 (2018)
- Jannesari, M., et al.: Breast cancer histopathological image classification: a deep learning approach. In: Proceedings of the 2018 IEEE International Conference on Bioinformatics and Biomedicine (BIBM), pp. 2405–2412 (2018)
- Xie, J., et al.: Deep learning based analysis of histopathological images of breast cancer. Front. Genet. 10, 80 (2019)
- Alom, M.Z., Yakopcic, C., Nasrin, M.S., Taha, T.M., Asari, V.K.: Breast cancer classification from histopathological images with inception recurrent residual con- volutional neural network. J. Digital Imaging 32(4), 605–617 (2019).
- Spanhol, F.A., et al.: A dataset for breast cancer histopathological image classifi- cation. IEEE Trans. Biomed. Eng. 63(7), 1455–1462 (2016)
- Shahidi, F., et al.: Breast cancer classification using deep learning approaches and histopathology image: a comparison study. IEEE Access 8, 187531–187552 (2020)
- Cohen, S.: Artificial Intelligence and Deep Learning in Pathology. Elsevier, Amsterdam (2020)
- Mormont, R., et al.: Comparison of deep transfer learning strategies for digital pathology. In: Proceedings of the 2018 IEEE Conference on Computer Vision and Pattern Recognition Workshops (CVPRW), pp. 2343–234309. IEEE (2018)
- Maree, R.: The need for careful data collection for pattern recognition in digital pathology. J. Pathol. Inform. 8(1), 19 (2017)
- Barisoni, L., et al.: Digital pathology and computational image analysis in nephropathology. Nat. Rev. Nephrol. 16, 669–685 (2020)
- Introduction could be shortened: I suggest to leave aside the first general observations on AI, Ml and DL, well-known and not needed in this context.
- Similarly, also the final section is rather naive, and could be improved by deepening the more scientifically oriented considerations.
- The list of results with reported performance figures is interesting but makes the manuscript hard to read - I would recommend maybe adding some tables to summarise the achieved results.
- It would be extremely useful, for at least a subset of the reviewed references, also describe and comment the different choices of DL architectures employed, also highlighting pros and cons of such structures.
- English syntax is non-standard in some paragraphs: a careful proofreading by a native speaker would be recommended/
- Finally, there are quite a few typos in the manuscript needed to be fixed (e.g. line 4 “Filipppo”, line 8 missing institution, line 97 “allowed to”, line 211 “Coundray”, line 281 “perspetives”, line 282 “ad vent’, …)
Overall, I would recommend a major review.
Author Response
Please see the attachement.

Round 2
Reviewer 2 Report
All the raised issues have been reasonably met.